# Cultural Adaptation and Validation of the Korean Version of the iMTA Productivity Cost Questionnaire

**DOI:** 10.3390/healthcare8020184

**Published:** 2020-06-24

**Authors:** Hyungtae Kim, Kyoung Sun Park, Jeong-Eun Yoo, Siin Kim, Sola Han, Hae Sun Suh

**Affiliations:** 1College of Pharmacy, Pusan National University, Busan 46241, Korea; jackja@pusan.ac.kr (H.K.); siin@pusan.ac.kr (S.K.); solasi@pusan.ac.kr (S.H.); 2Jaseng Spine and Joint Research Institute, Jaseng Medical Foundation, Seoul 06110, Korea; lovepks0116@daum.net; 3Department of Korean Medicine Obstetrics and Gynecology, Dunsan Korean Medicine Hospital of Daejeon University, Daejoen 35235, Korea; koreadryoo@gmail.com

**Keywords:** productivity losses, absenteeism, presenteeism, unpaid work, cultural adaptation, validation

## Abstract

This study develops the Korean version of the Institute for Medical Technology Assessment Productivity Cost Questionnaire (iPCQ) through translation/cultural adaptation and evaluation of psychometric properties. We included 110 outpatients visiting a gynecology clinic. We conducted the translation and cross-cultural adaptation of the iPCQ, including forward and back-translation, pilot test with cognitive debriefing, and finalization. We analyzed the feasibility (using average time of filling in the iPCQ and the proportion of missing values), test–retest reliability (using the intra-class correlation coefficient [ICC]), and validity (concurrent validity with the Work Productivity and Activity Impairment [WPAI] and construct validity with the 36-Item Short Form Survey [SF-36], using Spearman’s ρ). The Korean version of iPCQ showed appropriate feasibility (average filling in time was 5.0 min without missing values), and had excellent values in the domains of absenteeism, presenteeism, and unpaid work for test–retest reliability (ICC: 0.92–0.99). For concurrent validity, the Korean version of iPCQ showed moderate–high correlation for absenteeism and presenteeism with the WPAI. All domains of productivity losses measured by the Korean version of iPCQ showed negative correlation with the quality of life estimated by the SF-36. Through this study, we developed a Korean instrument that can measure and value health-related productivity losses including unpaid work as well as absenteeism and presenteeism.

## 1. Introduction

According to a recent study, in Korea, the economic burden of disease in 2015 was USD 133.7 billion, of which the cost of productivity loss was USD 68.2 billion [1]. This study showed that disease burden was a large part of the cost of productivity loss. The impact of productivity loss on disease burden was high and important [2]. Cost of productivity loss is recommended to be considered in health-related economic evaluation [3,4].

Costs of health-related productivity loss can be defined as costs associated with production loss and replacement costs due to illness, disability, and death of productive persons, both paid and unpaid [5]. Productivity loss comprises absenteeism, presenteeism, and unpaid work [6]. Guidelines and reports on health-related economic evaluation recommend measuring the cost of productivity loss using appropriate instruments [7,8,9]. Several instruments have been developed to measure and value health-related productivity loss [7,8,10,11,12]. Among these, the major instrument used to measure productivity loss in Korea is the Work Productivity and Activity Impairment (WPAI) questionnaire. This instrument measures all components of productivity loss including absenteeism, presenteeism, and unpaid work. However, it is difficult to convert the results of unpaid work to monetary value [11]. Recently, the Institute for Medical Technology Assessment (iMTA) Productivity Cost Questionnaire (iPCQ) has been developed to measure and value health-related productivity loss including absenteeism, presenteeism, and unpaid work [13]. The objective of this study was to adapt the iPCQ into the Korean language for cross-cultural suitability and to evaluate the psychometric properties of the Korean version of the iPCQ.

## 2. Materials and Methods

### 2.1. iPCQ and Other Instruments

The iPCQ has five domains with 18 items. Among these, the domain of general questions regarding the target population (Item A1–Item A6) is used to estimate the characteristics of a patient. The other domains (including general questions regarding paid work [Item 1–Item 3], absenteeism module [Item 4–Item 6], presenteeism module [Item 7–Item 9], and unpaid work module [Item 10–Item 12]) are required to identify and value the loss in productivity [13].

The general questions regarding the target population identify the respondent’s characteristics including survey date, age, sex, level of education, and type of work. Researchers can perform subgroup analyses based on this domain. The general questions regarding paid work survey the characteristics of paid work, including work periods (in hours and in days) per week. Based on the answers, researchers can calculate the average work hours per day. The domains of each module (absenteeism, presenteeism, and unpaid work) comprise items that measure the characteristics of productivity loss (whether productivity loss occurs and the period of productivity loss for each domain). In addition, each module has distinct questions as follows: in the absenteeism module, there is a question to identify whether the absenteeism is short-term or long-term (Item 5). In the presenteeism module, Item 9 asks how much the productivity efficiency of paid work has been reduced compared to that under normal conditions, by measuring the extent of health-related problems on a scale of 0 to 10 points. In the module for unpaid work, item 12 measures how health-related problems have reduced the productivity of unpaid work, based on a “third-person criterion.” [7,13,14].

### 2.2. Study Design and Process

We conducted the translation and cross-cultural adaptation of the iPCQ into Korean (including forward translation, back-translation, pilot test and cognitive debriefing, and finalization). Next, we analyzed the validity and reliability of the Korean version of iPCQ, including feasibility, concurrent validity, construct validity, and test–retest reliability.

To perform the validation study, we conducted surveys including the WPAI and the Short Form 36-Item health survey (SF-36), as well as the iPCQ. The WPAI is the most frequently used instrument for measuring productivity loss in Korea. This instrument comprises six items (currently employed [Item 1], hours missed due to health problems [Item 2], hours missed because of other reasons [Item 3], hours actually worked [Item 4], degree to which health affected productivity while working [Item 5], and degree to which health affected unpaid activities [Item 6]) and can estimate four main scores (including percent work time missed due to health, percent impairment while working due to health, percent overall work impairment due to health, and percent unpaid work impairment due to health) [15]. The SF-36 is an instrument designed to identify a patient’s quality of life and has been culturally adapted into Korean and validated [16]. This instrument includes 36 items creating 8 scales (physical functioning, role-physical, bodily pain, general health, vitality, social functioning, role-emotional, and mental health) [17].

### 2.3. Participants

We included outpatients (≥19 years old) who visited a gynecology clinic from August to September 2018 and agreed to participate. Based on a 5:1 subject-to-item rule (the Korean version of iPCQ has 18 items), at least 90 patients were required to conduct a validation study of the instrument [18]. Considering a dropout rate of 5%, approximately 95 patients were required. Additionally, we planned to recruit 15 patients to conduct a pilot test. Therefore, we included 110 patients in this study. This study was approved by the Institutional Review Board of the Korean Medicine Hospital of Daejeon University (DJDSKH-17-BM-34) and the KyungHee University Korean Medicine Hospital at Gangdong (KHNMCOH-2017-11).

### 2.4. Translation and Cultural Adaptation

We developed the Korean version of the iPCQ based on the guidelines for cross-cultural adaptation, including the process of preparation, forward translation, back translation, pilot test, cognitive debriefing, and finalization [19,20]. In the preparation step, we contacted the iPCQ development team (Institute for Medical Technology Assessment, iMTA) to grant permission for development of the Korean version of iPCQ through the iMTA website (https://www.imta.nl/questionnaires/). We explained the goal of our research and source of funding to the development team. We acquired the approval in May 2017.

In the forward translation, two independent translators, who were native Koreans with a good command of English, conducted forward translations of the original iPCQ into Korean. Our research team (H.K., S.H., S.K., and H.S.S.) reconciled the two forward translated Korean versions of the iPCQ into a single Korean version considering conceptual equivalence and cultural appropriateness.

In the back translation, two translators who are bilingual in English and Korean, without knowledge of the original iPCQ, received the reconciled forward-translated Korean version and independently conducted back translations. Our research team (H.K. and H.S.S.) reviewed and compared the conceptual equivalence and cultural appropriateness between the original version and the back translations. We repeated the discussion process until a consensus on the translation had been achieved, and then prepared the intermediate Korean version of the iPCQ.

We conducted a pilot test using the intermediate Korean version of the iPCQ with 15 outpatients (≥19 years) visiting gynecology clinics (7 patients in the Korean Medicine Hospital of Daejeon University, and 8 patients in the KyungHee University Korean Medicine Hospital). In the pilot test, we distributed the intermediate Korean version of the iPCQ to these patients and conducted cognitive debriefing. At the end of the questionnaire, we asked the patients the following questions: 1. “Was this question difficult to answer?”; 2. “Was this question confusing?”; 3. “Was this question difficult to understand?”; 4. “Was this question upsetting or offensive?”.

For finalization, a research member (H.K.) prepared a record sheet including the process of the five translations (two forward-translated versions, one reconciled version, and two back-translated versions) and the results of the cognitive debriefing. The research team (H.K. and H.S.S.) checked all issues and changes from the original to the translations and patients’ comments. We repeatedly reviewed and discussed the cultural appropriateness, conceptual equivalence, and difficulty of the translations until arriving at a consensus. Then, we developed the final Korean version of the iPCQ. We sent the final translations to the iMTA development team for final approval. After minor modifications of the instrument name and acknowledgement, we received approval for the Korean version of iPCQ.

### 2.5. Validation and Statistical Analysis

We conducted the validation study with outpatients (≥19 years old) who visited a gynecology clinic from August to September 2018. Before patients received medical care, we conducted surveys on the socio-economic characteristics, iPCQ, SF-36, and WPAI. We estimated the feasibility, including the mean respondent time spent filling out the questionnaire and the proportion of missing values for the Korean version of iPCQ [14]. We administered the iPCQ again after patients received medical care for the test–retest reliability.

We used the test–retest reliability measure to identify the extent to which the scores of the Korean version of iPCQ are similar at different times (2-h gap between before and after receiving medical care). We used the intra-class correlation coefficient (ICC) considering the following cut-off values: <0.5 (poor), ≥0.5 (moderate), ≥0.75 (good), ≥0.9 (excellent) [21].

We conducted the concurrent validity test to analyze the extent to which the scores of the Korean version of iPCQ correlate with an external criterion [22,23]. We used the Korean version of the WPAI:GH, which is a validated and widely used instrument in Korea, as the external criterion [15,24]. First, we divided the measurement domains into absenteeism, presenteeism, and unpaid work by referring to the manual for each instrument of the iPCQ and WPAI. Next, we measured the time related to productivity loss in each domain based on the results of the survey. Finally, we conducted a correlation analysis for each domain (absenteeism, presenteeism, and unpaid work) between the two instruments. Because all the variables we compared were not normally distributed, we used the Spearman’s rank correlation coefficient ρ (|ρ| < 0.3: low, 0.3 ≤ |ρ| < 0.6: moderate, 0.6 ≤ |ρ|: high) [25].

We performed the construct validity to evaluate whether the productivity loss measured by the Korean version of iPCQ is correlated with the SF-36 domains (including physical functioning, role physical, bodily pain, general health, vitality, social functioning, role emotional, mental health) using the Spearman’s rank correlation coefficient ρ. We hypothesized that each domain of productivity loss (absenteeism, presenteeism, and unpaid work) estimated by the iPCQ was negatively correlated with the quality of life. In general, it is known that a patient’s health problem has an effect on the measurement of the health-related productivity loss [25,26,27,28,29]. We classified the correlations into the follow categories: high correlation (r ≥ |0.3|) and low correlation (r < |0.3|) [27,28]. We used STATA version 15.0 (StataCorp, LP, College Station, TX, USA) for calculating the ICC. For the other analyses, we used SAS version 9.4 (SAS Institute, Cary, NC, USA).

## 3. Results

### 3.1. Translation and Cross-Cultural Adaptation of the iPCQ

Some issues were discussed in detail during the translation and cross-cultural adaptation process, including the cognitive debriefing. We modified the structure, explanation, and questions of the iPCQ considering the law and culture in Korea (Table 1).

### 3.2. Patient Characteristics

A total of 95 outpatients participated in the validation study, among whom two were excluded (one was under the age of 19 and the other had a missing value in the SF-36). The mean age of the 93 participants was 39.52 (standard deviation: 10.39). Table 2 shows the characteristics of the final respondents.

### 3.3. Feasibility and Reliability

The average time spent in filling out the Korean version of the iPCQ was 5.0 min with a standard deviation of 3.9 min. The percentage of missing values was 0%, as all respondents who participated in the study answered all questions appropriately.

For the test–retest reliability, the Korean version of iPCQ had excellent values for the productivity loss time of absenteeism, presenteeism, and unpaid work (ICC: 0.92–0.99) (Table 3).

### 3.4. Concurrent Validity between iPCQ and WPAI

Our study showed that a significantly high correlation for absenteeism was observed between the Korean version of iPCQ and WPAI:GH (Spearman’s ρ = 0.738), and correlation for presenteeism between these instruments was moderate (Spearman’s ρ = 0.483). However, the correlation between the unpaid work domain of the two instruments was low. Despite the correlations between the different domains of the two instruments being low or insignificant, the correlation between the absenteeism of iPCQ and the unpaid work of WPAI was moderate (Spearman’s ρ = 0.429) (Table 4).

### 3.5. Construct Validity between iPCQ and SF-36

Table 5 shows the results of the construct validity of the Korean version of iPCQ with the quality of life estimated by the SF-36. The period of absenteeism had negative and high Spearman’s correlation coefficient values with physical functioning, role physical, bodily pain, general health, vitality, and mental health (−0.46–−0.34). Presenteeism estimated by the Korean version of iPCQ was significantly correlated with the role physical, bodily pain, and social functioning (−0.45–−0.39). The period of productivity loss for unpaid work was negatively correlated with bodily pain (−0.32) and vitality (−0.36).

## 4. Discussion

This is the first study performing a cross-cultural translation and validation of the iPCQ in an Asian country. To the best of our knowledge, the WPAI is the only validated instrument that measures the health-related productivity loss—including absenteeism, presenteeism, and unpaid work—in Korea. However, the WPAI cannot calculate the monetary value of productivity loss in unpaid work. The iPCQ can value the productivity loss of unpaid work based on the “third-person criterion.” [7]. Through this study, we developed a Korean instrument that can measure and value health-related productivity loss, including unpaid work as well as absenteeism and presenteeism.

We recruited 93 participants in the validation stage of our study. Among these, 25.8% were students, housewives, and unemployed without paid work. We tried to adapt cultural bias and measure validity and reliability of the domain for the unpaid work by including participants without paid work. Previous study on the development of iPCQ also emphasized the importance of measuring the productivity loss of unpaid work, and included 38% of the population without paid work for the feasibility analysis [13].

Our study showed that the Korean version of iPCQ had appropriate feasibility and good reliability. The average time of filling out the Korean version of iPCQ was 5 min without missing values, which is similar to that in the previous study, wherein the original version of the iPCQ was developed [13]. The reliability in all domains of the Korean version of iPCQ was excellent (ICC > 0.9), and these results were similar to those reported in the study by Munk et al. [25].

In this study, we showed that the Korean version of iPCQ demonstrates moderate to high validity. Concurrent validity showed high correlation for absenteeism between the Korean version of iPCQ and WPAI:GH. However, the correlation for presenteeism between the Korean version of iPCQ and WPAI was moderate. This result might be owing to the different characteristics of the two instruments. The iPCQ measures the time and efficiency of presenteeism during the 4-weeks recall period [13]. In contrast, the WPAI assumes that the time of presenteeism is the remaining work time excluding the time of absenteeism during the 1-week recall period, and estimates the labor efficiency for this period. For the domain of unpaid work, there was no significant correlation between the two instruments. However, it is difficult to compare the results for unpaid work directly because each instrument identifies different properties of unpaid work. The iPCQ measures the “time” of productivity loss for unpaid work that has productivity, including household work and volunteer work, whereas the WPAI estimates the “proportion” of loss for every unpaid activity that does not consider productivity [8]. Additional studies are required to examine the validity of questions that measure and value the productivity loss of unpaid work.

For construct validity, as expected, the absenteeism, presenteeism, and unpaid work of the iPCQ showed negative correlation with the quality of life estimated by the SF-36. The domain of bodily pain, especially, showed a negative and significant correlation with all domains of productivity loss. Previous studies of various instruments to measure productivity loss have shown that the quality of life correlates negatively with productivity loss [27,28,29]. The results of our study are consistent with those of these reports.

Despite the good and appropriate feasibility, reliability, and validity of the Korean version of iPCQ, our study has some limitations. First, all participants in this study were women. To better measure the productivity loss of paid and unpaid work, we employed convenience sampling of female patients who participated in the randomized clinical trial for menstruation. In the process of cultural adaptation, there were no questions and answers which can be interpreted differently or sensitive by sex. To the best of our knowledge, no research has been conducted on the effect of sex on the validity of instruments of health-related productivity loss. However, caution is needed in the application of the results of this study to the general population, because the relationship between sex and health status cannot be excluded. Further studies are warranted to evaluate the validation of the Korean version of iPCQ among the general population including male. Second, we evaluated the test–retest reliability using two assessments within the same day. This may overestimate the reliability. However, we should always consider a trade-off in test–retest reliability between the potential effect of learning in a short-time interval, and the probability of change in a patient’s status during a long-term interval. In a previous study comparing the reliability of health-related instruments between short and long-term intervals, no significant difference was reported [30]. To avoid the fluctuations in productivity loss between different days, we conducted the Korean version of iPCQ twice on the same day to measure reliability.

## 5. Conclusions

In this study, the Korean version of iPCQ was translated and adapted according to the guidelines for cross-cultural adaptation. Our validation study showed that the Korean version of iPCQ had appropriate and good feasibility, reliability, and validity. The Korean version of iPCQ can be used to measure and value productivity loss, including absenteeism, presenteeism, and unpaid work in Korea.

## Figures and Tables

**Table 1 healthcare-08-00184-t001:** Issues and changes in the instrument.

Original Version	Issues	Changes in the Final Korean Version
Anonymity statement in the explanation on page 2–3	There is a law on the obligations to and protection of survey respondents, and many Korean surveys describe this.	Added the legal statement in Korean that anonymity will be ensured.
Item A4 to identify the patient’s level of education on page 4	The list of categorical answers does not fit the Korean education system.	Modified to consider the Korean education system using the standard levels used in the National Survey in Korea. ^(1)^
One of the categorical answers for item A5 “I am unable to work, for …%” on page 4	Answering this was regarded as difficult by the respondents as it is subjective.	Modified to consider the Korean system using categories used in the National Survey in Korea. ^(1)^
The statement regarding use of an envelope to send the completed questionnaire	Surveys are rarely conducted through postal services in Korea	Removed this statement.
Structure of the questionnaire: - Items 1–4 on page 6 - Items 5–8 on page 7 - Items 9–10 and explanation of the concept of unpaid work on page 8 - Items 11–12 and space for any comments from the respondent on page 9	Respondents were confused because items belonging to different modules were located on the same page or items belonging to one module were located on different pages.	Changed the location of each item considering the modules (absenteeism, presenteeism, and unpaid work). - Items 1–3 (for the general questions about paid work module) on page 6 - Items 4–6 (for the absenteeism module) on page 7 - Items 7–9 (for the presenteeism module) on page 8 - Explanation of the concept of unpaid work and items 10–12 (for the unpaid work module) on page 9 - Space for any comments from the respondent on page 10
Phrases instructing item movement	Too many statements between questions caused confusion in the respondents. In general, we included numbers to go to after each answer option in Korean. Thus, respondents were able to fill the questionnaire without missing questions.	Changed the location of the phrases from “between questions” to “just after each answer option”
Log of questions for measuring short episodes of absence from work	Not considered in this study	Not included in this study

^(1)^ The Korea National Health and Nutritional Examination Survey.

**Table 2 healthcare-08-00184-t002:** Characteristics of the respondents (Number of female: 93).

Characteristics	Number (%)	
**Total patients**	93	(100.00)
**Age (years)**		
20–29	16	(17.20)
30–39	32	(34.41)
40–49	29	(31.18)
50–59	14	(15.05)
Over 60	3	(3.23)
**Sex (Female)**	93	(100.00)
**Education**		
Elementary school	1	(1.08)
Middle school	1	(1.08)
High school	18	(19.35)
College or University	55	(59.14)
Graduate or higher	18	(19.35)
**Marital status**		
Married	71	(76.34)
Divorced/widowed	5	(5.38)
Single	17	(18.28)
**Type of work**		
Student	10	(10.75)
Employed or self-employed	69	(74.19)
Housewife	11	(11.83)
Unemployed	3	(1.08)
Retiree	0	(0.00)

**Table 3 healthcare-08-00184-t003:** Test–retest reliability results.

Analysis	The Korean Version of iPCQ		
	Absenteeism Time Per Month ^(1)^	Presenteeism Time Per Month × Amount of Work Efficiency ^(2)^	Time of Productivity Loss in Unpaid Work ^(3)^
Intra-class correlation coefficient	0.95 **	0.92 **	0.99 **

iPCQ, the Institute for Medical Technology Assessment Productivity Cost Questionnaire ^(1)^ (Item2_iPCQ_/Item3_iPCQ_) × Item4_iPCQ_; ^(2)^ (Item2_iPCQ_/Item3_iPCQ_) × Item8_iPCQ_ × [(10 - Item9_iPCQ_)/10]; ^(3)^ Item11_iPCQ_ × Item12_iPCQ_; ** *p*-value < 0.001.

**Table 4 healthcare-08-00184-t004:** Concurrent validity results using the Spearman’s rank correlation.

WPAI	The Korean Version of iPCQ		
	Absenteeism Time Per Month ^(1)^	Presenteeism Time Per Month × Amount of Work Efficiency ^(2)^	Time of Productivity Loss in Unpaid Work ^(3)^
**[Absenteeism time per week] ^(4)^**	0.738 **	−0.198	0.062
**[Presenteeism time] × [Degree health affected productivity while working] ^(5)^**	0.029	0.483 *	0.183
**[Degree of unpaid work-impairment due to health] ^(6)^**	0.429 *	0.092	0.185

iPCQ, the Institute for Medical Technology Assessment Productivity Cost Questionnaire; WPAI, Work Productivity and Activity Impairment Questionnaire ^(1)^ (Item2_iPCQ_/Item3_iPCQ_) × Item4_iPCQ_; ^(2)^ (Item2_iPCQ_/Item3_iPCQ_) × Item8_iPCQ_ × [(10 - Item9_iPCQ_)/10]; ^(3)^ Item11_iPCQ_ × Item12_iPCQ_; ^(4)^ Item2_WPAI_; ^(5)^ Item4_WPAI_ × (Item5_WPAI_/10); ^(6)^ Item6_WPAI_/10; * *p*-value < 0.05, ** *p*-value < 0.001.

**Table 5 healthcare-08-00184-t005:** Construct validity results using Spearman’s rank correlation.

SF-36 (Score)	The Korean Version of iPCQ		
	Absenteeism Time Per Month ^(1)^	Presenteeism Time Per Month × Amount of Work Efficiency ^(2)^	Time of Productivity Loss in Unpaid Work ^(3)^
Physical Functioning	−0.36 *	−0.10	−0.21
Role Physical	−0.37 *	−0.39 *	−0.20
Bodily Pain	−0.46 **	−0.45 *	−0.32 *
General Health	−0.38 *	−0.28	−0.20
Vitality	−0.38 *	−0.20	−0.36 *
Social Functioning	−0.27 *	−0.52 *	−0.14
Role Emotional	−0.26	−0.36	−0.05
Mental Health	−0.34 *	−0.15	−0.22

iPCQ, the Institute for Medical Technology Assessment Productivity Cost Questionnaire; SF-36, Short Form 36-Item health survey ^(1)^ (Item2_iPCQ_/Item3_iPCQ_) × Item4_iPCQ_; ^(2)^ (Item2_iPCQ_/Item3_iPCQ_) × Item8_iPCQ_ × [(10 - Item9_iPCQ_)/10]; ^(3)^ Item11_iPCQ_ × Item12_iPCQ_ * *p*-value < 0.05, ** *p*-value < 0.001.

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
