# Peer review of "Cultural Adaptation and Validation of the Korean Version of the iMTA Productivity Cost Questionnaire"

_healthcare, 2020, doi:10.3390/healthcare8020184_

Round 1
Reviewer 1 Report
1.The subjects include 10 students in this study,and how to calculate the costs of health-related productivity loss in these students? and have any bias to the study findings?
2.following the above question, and the unemployed and housewife , have also the same problem, too.
3. as table 2 show all participants in this study were women, and why the same table show housewife/househusband?
4. the topic state its a cultural adaption, why the research subjects are single gender oriented? and it also choose the gynecology clinic as the study setting. It is very hard to assess the purpose of this designed scale as the following application.
Reviewer 2 Report
It seems to me that it is well designed and executed and that it is relevant to be published, as it can serve as a model for validation in other countries. I think that it should only be improved regarding table 2, in which there is redundant information, with putting in the title of the table that there are 93 women, the corresponding row can be eliminated, later in age, if the important thing is set the age groups, putting the mean age and standard deviation in the text could eliminate that row and finally, if the absolute frequency (%) is put in the column heading, it is no longer necessary to include the% symbol in all% of rows.
I think that by modifying those details, the article can be published
Reviewer 3 Report
The article " Cultural adaptation and validation of the Korean version of the iMTA Productivity Cost Questionnaire ", is an original research, and it can be publishable in the Journal.
Very well prepared article. Based on quantitative research. Research carried out correctly. The presented results are very interesting for readers.
I congratulate the authors for the research work done
Round 2
Reviewer 1 Report
Revised appropriately and can be accepted.